# Vitamin D and Dyslipidemia: Is There Really a Link? A Narrative Review

**DOI:** 10.3390/nu16081144

**Published:** 2024-04-12

**Authors:** Antonella Al Refaie, Leonardo Baldassini, Caterina Mondillo, Michela De Vita, Elisa Giglio, Roberto Tarquini, Stefano Gonnelli, Carla Caffarelli

**Affiliations:** 1Section of Internal Medicine, Department of Medicine, Surgery and Neuroscience, University of Siena, 53100 Siena, Italygonnelli@unisi.it (S.G.);; 2Division of Internal Medicine I, San Giuseppe Hospital, 50053 Tuscany, Italy

**Keywords:** vitamin D, dyslipidemia, cardiovascular prevention, cholesterol, triglycerides

## Abstract

Nowadays, the interest in the extraskeletal effects of vitamin D is growing. In the literature, its several possible actions have been confirmed. Vitamin D seems to have a regulatory role in many different fields—inflammation, immunity, and the endocrine system—and many studies would demonstrate a possible correlation between vitamin D and cardiovascular disease. In this paper, we deepened the relationship between vitamin D and dyslipidemia by reviewing the available literature. The results are not entirely clear-cut: on the one hand, numerous observational studies suggest a link between higher serum vitamin D levels and a beneficial lipid profile, while on the other hand, interventional studies do not demonstrate a significant effect. Understanding the possible relationship between vitamin D and dyslipidemia may represent a turning point: another link between vitamin D and the cardiovascular system.

## 1. Introduction

Dyslipidemia is an imbalance of lipids such as total cholesterol (TC), low-density lipoprotein cholesterol (LDL-C), triglycerides (TGs), and high-density lipoprotein (HDL-C). It is a pathological condition found all around the world. Dyslipidemia is also a fundamental and modifiable cardiovascular risk factor. Genetic and environmental factors (obesity, incorrect diet, smoking) are both involved. Excessive cholesterol poses a risk, but it is essential in appropriate amounts for the human body; in fact, it maintains the integrity and fluidity of cell membranes and represents the precursor of hormones and bile acids.

Vitamin D undergoes an initial conversion to 25-hydroxyvitamin D (25OHD) in the liver, followed by a subsequent transformation into its active form, 1,25-dihydroxyvitamin D, or calcitriol, primarily occurring in the kidneys and various tissues. The synthesis of 1,25-dihydroxyvitamin D in the kidneys is regulated by levels of parathyroid hormone, as well as concentrations of calcium and phosphorus in the bloodstream. Vitamin D3, also referred to as cholecalciferol, shares structural similarities with steroids and is generated in the skin upon exposure to ultraviolet light from the sun. Circulating 25(OH)D is the most dependable indicator of vitamin D status in humans, reflecting dietary intake, supplements, and the skin synthesis of vitamin D. Moreover, 25OHD’s functions do not stop only at the bone but involve the inflammatory, immune, and endocrine systems. Vitamin D deficiency is very common across numerous regions globally. In a recent study, the association between vitamin D deficiency and cardiovascular disease has been described [1]. 

How can vitamin D affect cholesterol levels? Actually, this is not entirely clear. Vitamin D and cholesterol share similarities in their biosynthesis processes: cholecalciferol is synthesized in the skin by UV-B irradiation from 7-dehydrocholesterol (7DHC); 7DHC is also converted to cholesterol by the action of the enzyme 7-Dehydrocholesterol reductase (DHCR7) [2]. Zou and Porter [3] demonstrated in vitro that increased levels of cholecalciferol lead to a rapid decrease in DHCR7 activity and result in a lipid decrease. This represents a fundamental connection between lipids and vitamin D (Figure 1). However, the relationship between vitamin D and lipids has deep-seated roots, extending to the genetic level. Below, we summarize the mechanisms that link vitamin D and lipids.

### 1.1. Genetic Mechanisms

Vitamin D is involved in lipid metabolism through genetic mechanisms mediated by the vitamin D receptor (VDR). The VDR gene is located on chromosome 12 (q11–q13). Several studies have shown that, in some populations, certain VDR polymorphisms are associated with higher levels of triglycerides and cholesterol [4,5,6]. Jia et al. [7] investigated the association between VDR polymorphism and dyslipidemia in a large Chinese population. Their work suggests that the presence of VDR rs2228570 polymorphism correlates with an increased risk of higher LDL-C and lower serum 25OHD levels in the study population. Additionally, it has been also proposed that VDR determines cholesterol levels by regulating the synthesis of bile acids at the genetic level: VDR inhibits liver X receptor alpha (LXR-alpha) signaling, thereby regulating bile acid and cholesterol homeostasis [8,9].

Li et al. demonstrated another important link between vitamin D and cholesterol trough VDR. Vitamin D deficiency reduces the transcriptional activity of VDR, leading to a down-regulation of insulin-induced gene-2 (Insig-2) expression and, consequently, the activation of its inhibitory effect on sterol regulatory element-binding protein 2 (SREBP-2). This cascade ultimately leads to an increase in 3-hydroxy-3-methylglutaryl-coenzyme A reductase expression, culminating in elevated cholesterol production [10]. 

### 1.2. Non-Genetic Mechanisms

There are also non-genetic mechanisms that link vitamin D and lipids (Figure 1). One of the main functions of vitamin D is to regulate calcium metabolism. Through this action, vitamin D interferes with lipid production in the following ways: (1) Increasing intestinal calcium absorption may play a role in modulating microsomal triglyceride transfer protein (MTP), thereby reducing the synthesis and secretion of triglycerides [11]. (2) Increasing intestinal calcium levels reduces the intestinal absorption of fatty acids due to the formation of insoluble calcium–fat complexes. (3) Calcium may promote the conversion of cholesterol into bile acids, resulting in decreased cholesterol levels. (4) 25OHD regulates PTH; a previous paper on rats demonstrated that hyperparathyroidism is associated with increased triglycerides. Therefore, 25-OH-vitamin D (25OHD), by regulating PTH, could regulate triglyceride values [12]. (5) Hypovitaminosis D could affect beta cell function and insulin resistance, thereby influencing lipoprotein metabolism and resulting in increased triglyceride levels and reduced HDL-C levels [13]. (6) Hypovitaminosis D is also associated with the pro-inflammatory state: inflammation, lipid metabolism, and atherosclerosis interact closely with each other [14]. (7) Finally, calcitriol (1,25(OH)D) seems to arrest cholesterol uptake by macrophages, thus suppressing foam cell formation, which is involved in atherosclerosis [15].

In this narrative review, we aim to delve deeper into the relationship between vitamin D and lipids to understand their potential link by analyzing observational and interventional studies.

## 2. Materials and Methods

A literature review was conducted from inception to July 2023. The PubMed/Medline, Cochrane Library, ClinicalTrials.gov, and SCOPUS databases were searched using the following search terms: “dyslipidemia” or “cholesterol” or “lipids” or “triglyceride” AND “vitamin D”. The process of selecting the studies for review in adherence to the PRISMA 2020 process is shown in Figure 2.

## 3. Results

### 3.1. Observational Studies

On this topic, it is possible to find observational and interventional studies. First, we began by analyzing observational studies. However, it is worth noting that interest in this topic is not recent. One of the earliest papers in 1992 was a Belgian study conducted on a group of 358 subjects. In this study, a blood sample was performed for the determination of total serum calcium, 25OHD, TC, HDL-C, apolipoprotein A-I (apoA1), apolipoprotein B (apoB), and total protein. The study determined the existence of a significant positive relation between serum 25OHD and both HDL-C and apo A-I [16]. It is well known that high HDL-C has a protective effect, unlike LDL-C and TGs. Cross-sectional studies with very large populations have reported an inversely significant correlation between the values of 25OHD, TGs, and LDL-C and a positive correlation with HDL-C, as reported by Aumerx et al. [16]. Similarly, Jiang too in 2018, in a cross-sectional study carried out on 3788 subjects, found an inversely significant correlation between 25OHD, TGs, and LDL-C, while there was a positive correlation with HDL-C [17]. These results further underline the possible protective role of vitamin D in reducing cardiovascular risk. Other studies confirmed the inverse correlation between 25OHD and TG or LDL-C serum levels [18,19,20]. In 2007, Botella-Carretero led a transversal observational study on 73 patients with morbid obesity. Anthropometric variables, 25OHD, lipid profiles (TC, HDL-C, LDL-C, TGs), glucose and insulin levels, and insulin resistance were measured for each patient. Vitamin D deficiency was present in 37 of 73 patients (50.7%) and was more prevalent in morbidly obese patients with metabolic syndrome. HDL-C was lower and TGs were higher in the hypovitaminosis D group [21]. Similar positive associations have been observed in numerous other studies conducted across various populations [22,23,24,25,26,27,28,29,30,31,32]. A recent cross-sectional study confirmed that the percentage of individuals with dyslipidemia and dysglycemia was higher in subjects with low levels of vitamin D [33]. Results from some studies suggested a possible correlation with gender. In a study by Wang et al. [34], vitamin D deficiency seemed to be associated with an increased risk of dyslipidemia more in men than in women. However, in a more recent paper by AlQuaiz et al., it was observed that hypovitaminosis D was associated with lower HDL-C levels in men and higher TG levels in women [35]. Many authors studied vitamin D and lipid profiles in young adults and children; also in these groups, vitamin D deficiency seemed to be associated with an increased risk of dyslipidemia [36,37,38,39,40,41,42,43,44,45]. A study by Al-Ajlan et al. analyzed the relationship between vitamin D and lipids in Saudi pregnant women, who had higher incidence of obesity, diabetes, and cardiovascular disease during the course of pregnancy. Surprisingly, serum vitamin D levels positively correlated with serum levels of TC and TGs [46]. Conversely, a prospective study carried out on a large cohort of Brazilian women reported that low 25OHD concentrations during early pregnancy were associated with higher TC, LDL-C, and TC/HDL-C ratios throughout pregnancy [47]. This correlation was recently confirmed in a study carried out on 2479 pregnant Chinese women in the second trimester. Serum 25OHD levels were inversely associated with TC, TG, HDL-C, LDL-C, and hs-CRP levels, suggesting that higher levels of vitamin D during pregnancy may improve lipid levels and inhibit the rise of hs-CRP [48].

The effect of vitamin D on lipid profile has also been studied in subpopulations with higher cardiovascular risk, such as patients with type 2 diabetes mellitus (T2DM), finding significant correlations [49,50,51,52]. Insulin resistance has effects on lipids [53]; therefore, vitamin D, having a regulatory role on insulin resistance [54], could also act on lipids in diabetic patients through this mechanism.

A meta-analysis conducted by Jafari et al. aimed to clarify the effect of vitamin D on serum TC, TGs, LDL-C, and HDL-C in T2DM patients; 17 studies comparing an intervention group (which received vitamin D) with a control group (which received a placebo) were enrolled in the study. The meta-analysis demonstrated that vitamin D improved serum levels of TC, TGs, and LDL-C in patients with T2DM, although changes in serum HDL-C levels were not significant [55].

One of the most influential studies on vitamin D and lipids was conducted by Ponda et al. in 2012 [56]. In this study, 4.06 million laboratory test results from September 2009 to February 2011 were analyzed. The study was both a cross-sectional study and a retrospective cohort analysis. While vitamin D deficiency was associated with an unfavorable lipid profile in cross-sectional analyses, replenishing vitamin D deficiency up to normal levels did not change lipid concentrations. Thus, although observational studies suggest that vitamin D deficiency impacts lipoprotein levels, Ponda’s study reported that this effect was not confirmed in interventional studies. Table 1 describes the main characteristics of observational studies.

### 3.2. Interventional Studies

In addition to observational studies on the link between vitamin D and lipids, several interventional studies on this topic are available in the literature. One of the earliest studies dates back to 1970 by Carlson et al., who examined the effect of vitamin D supplementation for 6 weeks on lipid profiles. The study concluded that replenishing vitamin D had no impact on lipid concentrations [57]. Yousefi Rad carried out a study on 58 diabetic patients, where for 2 months, 28 subjects received 100 micrograms (4000 IU) of vitamin D while 30 subjects received a placebo. HDL-C levels increased significantly in both groups; no difference was observed in the replenishment group [58]. A double-blind randomized placebo-controlled trial on 70 diabetic patients, who received vitamin D or a placebo for 12 weeks, showed a significant reduction in TC, LDL-C, and TGs, in both groups. HDL-C level decreased significantly only in the placebo group; however, the variables did not all achieve statistical significance [59]. Moreover, numerous studies over time have shown that there is no change in TC, LDL-C, HDL-C, and TGs after treatment with different doses of vitamin D [60,61,62,63,64,65,66,67,68,69,70,71,72,73,74,75,76,77,78,79,80,81,82,83,84,85,86,87,88,89,90]. A prospective, randomized, open-label trial carried out on healthy Indian men compared the effect of increased sunlight exposure versus vitamin D supplementation on 25OHD status and lipid profile in subjects with vitamin D deficiency. In the sunlight exposure group, an increase in vitamin D levels significantly reduced TC, LDL-C, and HDL-C concentrations, while in the cholecalciferol supplementation group, TC and HDL-C levels increased [91]. However, these data are not unique. In fact, there are also studies which have demonstrated that vitamin D supplementation can have effects on lipids. This is the case of the paper of Farag et al. [92], in which daily vitamin D supplementation for 12 weeks associated with moderate physical activity led to a significant reduction in lipid profile in patients with metabolic syndrome. In the paper of Salekzamani [93], vitamin D replenishment seemed to have positive effects only on TGs. Also, Asemi et al. demonstrated a significant decrease in TGs and VLDL-C in a population who received calcium and vitamin D [94]. This reduction in TG levels was also observed in other studies [95,96,97,98,99,100].

In the study by Imga et al., carried out on 122 overweight or obese women who underwent to vitamin D replenishment, it was concluded that vitamin D supplementation led to an improvement in HOMA-IR and LDL-C values [101]. The same results were demonstrated in a group of 120 adult Saudi patients with controlled T2DM who received 2000 IU of vitamin D daily for 18 months. The vitamin D replenishment brought about an improved lipid profile (a decrease in LDL-C and TC) and an improvement in HOMA-β function [102]. Shab-Bidar et al., in a study carried out into T2DM, found a decrease in LDL-C and TC along with a significative increase in HDL-C after vitamin D supplementation [103]. Replenishment with vitamin D was also associated with decreased TC in a T2DM Spanish study [104]. A recent study by Sacheck JM et al. found that replenishment with vitamin D had positive effects on HDL-C, LDL-C, and TC [105] in young people. Significant improvements in serum HDL-C in a vitamin D group was also found by Liyanage [106] in subjects with T2DM and early-stage nephropathy. An Iranian randomized controlled trial carried out on 168 subjects with prediabetes showed significantly increased HDL-C levels in the co-supplementation group (with omega 3 and vitamin D) compared to the other three groups (placebo group, only omega 3 group, only vitamin D group) [107]. Samimi et al. carried out a double-blind, placebo-controlled trial on 60 women at risk for pre-eclampsia from 20 to 30 weeks of gestation; in particular, 30 women received 50,000 IU of cholecalciferol every 2 weeks plus 1000 mg of calcium per day, and 30 women received a placebo [108]. The study demonstrated that vitamin D plus calcium administration for 12 weeks had beneficial effects on HDL-C, glycemic status, and blood pressure among women at risk for pre-eclampsia. In 2018, Riek et al. studied the potential role of vitamin D in cardiovascular risk in diabetic patients, with very interesting results [109]. This study was based on the fact that vitamin D and its metabolites have effects on atherogenesis, acting on macrophages and peripheral blood monocytes in patients with diabetes [110]. Riek randomized 26 diabetic patients to receive either vitamin D at a dosage of 4000 IU/day or a placebo for 4 months. Monocytes of the patients were isolated; in 2TDM patients who received vitamin D, levels of oxidized LDL uptake in cultured monocytes had a reduction >50%. These data would support the positive effect of vitamin D on atherosclerosis and lipid carriers [110]. Moreover, in a group of 181 subjects with mild cognitive impairment, a replenishment for 12 months with 400 UI/day of vitamin D showed decreased levels of TC, TGs, and LDL-C as well as increased HDL-C [111]. In a very recent interventional study, the correction of vitamin D deficiency in Arab adults seemed to improve their 10-year risk of atherosclerotic cardiovascular disease (ASCVD). This result is very interesting; once optimal vitamin D levels were achieved, there was a modest improvement in glucose and HDL-C levels, resulting in a decrease in ASCVD risk scores [112]. However, a study by Schwetz et al. demonstrated that vitamin D supplementation significantly increased TC, TGs, VLDL-C, LDL-C, HDL-C, ApoB, and ApoB, indicating a negative effect on lipid profile [113]. An increase in LDL-C levels after replenishment was observed also by Zittermann in 2009 [114]. The non-univocal interpretation of the results on the relationship between vitamin D and lipids is evident in the studies of Jorde. In a cross-sectional study from 2010 carried out on 1762 nonsmoking and 397 smoking subjects, Jorde [115] found a significative decrease in TGs with an increase in vitamin D. In the same year, Jorde randomized 438 overweight or obese subjects to receive (a) 40,000 IU of vitamin D weekly, (b) 20,000 IU of vitamin D weekly, or (c) a placebo for one year; the result was that replenishment with vitamin D had no effect on serum lipid levels. In 2016, in a randomized placebo-controlled trial conducted on 511 subjects with prediabetes, Jorde also found a significant decrease in LDL-C in the vitamin D supplementation group [116]. Moreover, the most recent papers show conflicting results on the effect of vitamin D supplementation on dyslipidemia [117,118,119]. Table 2 describes the main characteristics of interventional studies.

In Figure 3, we report the number of studies presenting responses to treatment with vitamin D.

Out of the 64 interventional studies examined, 35 (55%) indicated no significant effects on lipid profiles following vitamin D supplementation, whereas 29 (45%) demonstrated a significant alteration in lipid profile. Taking into consideration the studies carried out on patients with T2DM, we have observed that treatment with vitamin D gave positive answers in a greater number of studies (57%). Also, in the group of studies conducted in patients with metabolic syndrome or who were overweight/obese, it was found that the number of studies with positive results was slightly higher than those with negative results. On the other hand, in studies conducted on healthy subjects, the response rate was markedly lower.

In Figure 4 we reported the effect of Vitamin D supplementation on lipid profile in different categories of patients.

**Figure 4 nutrients-16-01144-f004:**
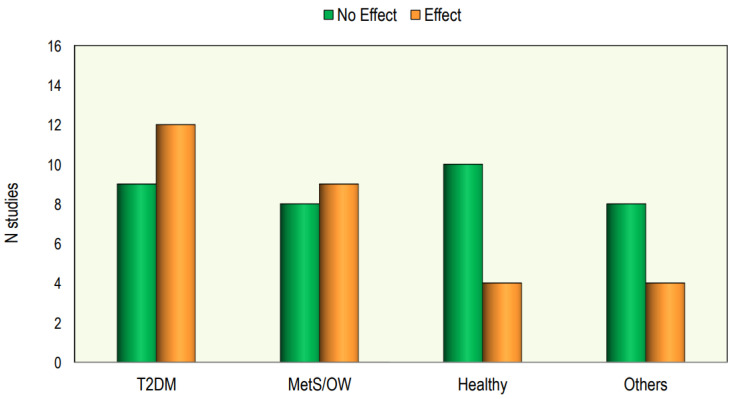
The effect of Vitamin D supplementation on lipid profile in different categories of patients.

**Table 1 nutrients-16-01144-t001:** Main characteristics of observational studies.

Study/Years	Country	Subjects	Study	Description/Evaluation	Results/Conclusion
Auwerx J et al. (1992) [16]	Belgium	185 ♂. Age: 38.7 ± 10.9 yrs 25OHD: 29.6 ± 12.1 ng/mL 173 ♀. Age: 37.2 ± 10.4 yrs 25OHD: 30.4 ± 14.5 ng/mL	Observational	total serum calcium, 25OHD, y-GT, TC, HDL-C, apo A-1, apo B, and total protein	Positive correlation between 25OHD and apoA-1 and HDL-C levels
Botella-CarreteroJI et al. (2007) [21]	Spain	73 obese patients 36 without vit D deficiency (Age: 42.17 ± 11.6 yrs 25OHD: 45.7 ± 35.5 ng/mL 37 with vit D deficiency (Age: 39.07 ± 12.7 yrs 25OHD 13.3 ± 3.8 ng/mL)	Transversal, observational	BMI, FG, 25OHD, lipid profile, glucose, insulin and insulin resistance	25OHD deficiency was more prevalent in morbidly obese patients with metabolic syndrome; HDL-C was lower and TGs were higher in the hypovitaminosis D group
Kostecka D et al. (2022) [23]	Poland	191 ♀ (45–65 yrs) 25OHD: 23 ng/mL	Observational	lipid profile, glycemia and 25OHD	Negative correlation between 25OHD and TC, LDL-C, and TGs
Chaudhuri J.R. et al. (2012) [18]	India	91 (48 ♂ and 43 ♀) normal vit D (Age: 49 ± 16.1 yrs) 91 (48 ♂ and 43 ♀) vit D deficiency (Age: 50.1 ± 15.1 yrs)	Observational	FG, lipid profile calcium, alkaline phosphatase, phosphorus, CRP, 25OHD	Hypovitaminosis D was associated with increased levels of TC, LDL-C and TGs and lower levels of HDL-C
Karhapää P. et al. (2010) [19]	Finland	909 ♂ DMT2 Age: 45–70 yrs 25OHD = 22.1–163.6 nmol/L	Observational	lipid profile, 25OHD, and 1,25OHD	Low levels of 1,25OHD were associated with low HDL-C levels; low levels of 25OHD were associated with high levels of TC, LDL-C and TGs
Guasch A. et al. (2012) [20]	Spain	76 ♂ Age: 49.28 yrs 25OHD: 65.01 nmol/L 240 ♀ Age: 46.08 yrs 25OHD: 53.55 nmol/L	Retrospective	FG, calcium, phosphate, alkaline phosphatase, lipid profile, creatinine, serum albumin, erythrocyte sedimentation rate (ESR) and leukocyte count, uCRP	Low levels of 25OHD were associated higher levels of TGs
Jiang X et al. (2019) [12]	China	3788 adults 2056 (54.28%) had dyslipidemia 25OHD: ≤8.02–≥33.71 nmol/L	Cross-sectional	lipid profile and 25OHD	25OHD was inversely correlated with LDL-C and TG levels, and positively correlated with HDL-C
Sharba ZF et al. (2020) [22]	Iraq	58 ♂ and 72 ♀. Age: 20–70 yrs 3 groups by serum level of vit D: <10 ng/mL—vit D deficient 10–30 ng/mL—vit D insufficient 30–100 ng/mL vit D normal	Cross-sectional	lipid profile and 25OHD	HDL-C was significantly reduced in low levels of 25OHD, while LDL-C and TG levels were increased
Saheb Sharif-Askari F S et al. (2020) [24]	Arab Emirates	1848 ♂, 641 ♀. Age: 18 to 80 yrs Insulin resistance group 25OHD: 28.50 ng/mL Insulin-sensitive group 25OHD: 31.20 ng/mL	Cross-sectional	lipid profile and 25OHD, IL-6, IL-8, and soluble thrombomodulin	Hypovitaminosis D was associated with lower HDL-C
Guan C. et al. (2020) [25]	China	10,038 subjects 25OHD: <20 ng/mL	Cross-sectional	lipid profile and 25OHD	Deficient serum 25OHD was associated with higher TC, LDL-C, and TGs
Han YY et al. (2021) [28]	China	715 (527 ♂ and 188 ♀) Age: 35–65 yrs first group: 25OHD < 15 ng/mL second group: 25OHD ≥15 ng/mL	Observational	lipid profile and 25OHD, glucose	Hypovitaminosis D was associated with increased levels of LDL-C, TGs and VLDL-C
Yang K. et al. (2020) [30]	China	1928 (958 ♂–970 ♀) Age: 18–87 yrs 25OHD: <15 nmol/L	Observational	lipid profile, 25OHD, FPG, H2PG, BMI	25OHD level was negatively correlated with FPG, TC and TGs
Wang Y. et al. (2016) [49]	China	1475 829 ♂ Age: 25–64.5 yrs 646 ♀ Age: 24–64 yrs 25OHD: 27–92.25 nmol/L	Cross-sectional	lipid profile and 25OHD, BMI	In ♂, elevated TGs and reduced HDL-C were associated with hypovitaminosis D; in ♀, no significant difference
AlQuaiz AM et al. (2020) [35]	Saudi Arabia	653 ♂ Age: 40.1 ± 10.2 yrs 1064 ♀ Age: 39.1 ± 8.3 yrs first group 25OHD: <50 nmol/L second group 25OHD: ≥50 nmol/L	Cross-sectional	lipid profile and 25OHD	Hypovitaminosis D was associated with low levels of HDL-C (higher in ♂ than in ♀) and high levels of TGs (in ♀ but not in ♂)
Kostrova GN. et al. (2022) [36]	Russia	64 boys and 214 girls Age: 18–24 yrs 25OHD: 14.9–26.3 ng/mL	Observational	lipid profile and 25OHD, BMI	In ♂, a negative correlation was found between 25OHD levels and TC and LDL
Kim MR. et al. (2019) [37]	Korea	117 boys and 126 girls Age: 9–18 yrs 25OHD: 17.27 ng/mL	Observational	lipid profile and 25OHD, BMI	The vitamin D-deficient group showed higher TG levels and TG/HDL-C ratios
Delvin EE et al. (2010) [38]	French Canadian	878 boys and 867 girls Age: 9–16 yrs Boys: 25OHD: 45.9 ± 12.2 nmol/L Girls: 25OHD: 45.9 ± 13.0 nmol/L	Cross-sectional	lipid profile and 25OHD, fasting plasma insulin, glucose, apolipoproteins (apo) A1 and B	Modestly higher concentration of plasma TC, TGs, apoA1, and apoB for each 10 nmol/L of 25OHD increase in plasma in girls only
Williams DM et al. (2011) [39]	USA	7078 subjects Age: 12–19 yrs 25OHD: 50.4 nmol/L	Cross-sectional	lipid profile and 25OHD, fasting insulin and glucose, post-load glucose and HbA1c	25OHD was positively associated with HDL-C values
Rajakumar K. et al. (2011) [40]	USA	237 subjects Age: 12.7 ± 2.2 yrs 25OHD: 19.4 ± 7.4 ng/mL	Observational	lipid profile and 25OHD, BMI	Lower levels of 25OHD are associated with lower HDL-C
Birken CS et al. (2015) [41]	Canada	996 boys and 965 girls Age: 1–5 yrs 25OHD: 85 nmol/l	Cross-sectional	lipid profile and 25OHD	A significant association between increased 25OHD and decreased non-HDL-C; each 10 nmol/L increase in 25OHD was associated with a decrease in non-fasting TC and in non-fasting TGs
Yarparvar, A. et al. (2020) [43]	Iran	71 boys (17 yrs) 25OHD: first group < 25 ng/mL second group ≥ 25 ng/mL	Observational	lipid profile and 25OHD, IL-10, IL-6, hsCRP, and TNFR-2	HDL-C level was lower in hypovitaminosis D
Saeidlou N. et al. (2017) [44]	Iran	541 subjects Age: 5–60 yrs; In winter: 25OHD: 45.8 ± 24.26 ng/mL; in summer: 25OHD: 55.24 ± 37.47 ng/mL	Cross-sectional	lipid profile and 25OHD and comparison of values between summer and winter	Comparing serum lipid levels in summer and in winter showed a significant difference in TC, LDL-C, and HDL-C, but no significant effect was found for TGs
Song K. et al. (2020) [45]	Korea	3183 subjects Age: 12–18 yrs 25OHD: 6.15 ng/mL	Cross-sectional	lipid profile and 25OHD	Vitamin D deficiency is related with low HDL-C levels
Al-Ajlan, A et al. (2015) [46]	Saudi Arabia	515 pregnant ♀ Age: 28.71 ± 6.07 yrs 25OHD: 24.42 ± 15.4 nmol/L	Cross-sectional	lipid profile and 25OHD	Serum vitamin D values correlated positively with serum levels of TC and TGs
Lepsch, J. et al. (2017) [47]	Brazil	194 pregnant ♀ Age: 26.7 ± 5.5 yrs first group 25OHD: ≥75 nmol/L second group25OHD: <75 nmol/L	Cross-sectional	lipid profile and 25OHD	Women with low levels of 25OHD had higher LDL-C than those with adequate concentrations
Jin D. et al. (2020) [48]	China	2479 pregnant ♀ Age: 29.3 ± 4.2 yrs 25OHD: 40.08 nmol/L	Observational	lipid profile and 25OHD hs-CRP	Increased serum 25OHD was significantly associated with decreasing TC, TGs, HDL-C, LDL-C, and hs-CRP levels
Wang L. et al. (2020) [49]	China	2659 (Age: 54–66 yrs) 849 T2DM (324 ♂, 525 ♀) 25OHD: 29.77 ng/mL 913 IFG (352 ♂, 561 ♀) 25OHD: 29.26 ng/mL 897 NGT (344 ♂, 553 ♀) 25OHD: 31.12 ng/mL	Case–control	lipid profile and 25OHD, HOMA-IR, BMI	Adequate vitamin D levels could reduce the risk of IFG and T2DM by reducing the lipid profile
Saedisomeolia, A. et al. (2014) [50]	Iran	T2DM 108 Age: 47.65 ± 12.08 yrs sufficient group 25OHD: ≥50 nmol/L deficiency group 25OHD: <50 nmol/L	Cross-sectional	lipid profile and 25OHD, calcium, phosphorus, PTH	Subjects with vit D deficiency had higher serum levels of TC, TGs, and LDL-C and lower levels of HDL-C compared to subjects with vit D sufficiency. Association was statistically significant only for TGs
Huang, Y. et al. (2013) [51]	China	T2DM 1326 ♂ Age: 47.6 ± 11.3 yrs T2DM 1326 ♀ Age: 49.4 ± 13.1 yrs 25OHD: 25.4 ± 6.5 ng/mL	Cross-sectional	lipid profile and 25OHD, LPL, FFAs, FG, fasting insulin, apoA and apoB	Serum 25OHD concentration was positively associated with LPL
Raheem M. et al. (2022) [52]	Iraq	47 T2DM subjects (Age: 35–64 yrs) 43 healthy (Age: 37–65 yrs) first group 25OHD: >22.5 ng/dL second group 25OHD: <22.5 ng/dL	Observational	lipid profile and 25OHD, HOMA-IR, HbA1c, FG	FG, HOMA-IR, TC and TGs were significantly elevated in T2DM compared to controls when the serum 25OHD was markedly low
Ponda M. et al. (2012) [56]	USA	107.811 subjects Retrospective 2332 subjects 25OHD: <20 ng/mL 6260 subjects 25OHD: 20–30 ng/mL	Cross-sectional Retrospective cohort	association between lipid profile and 25OHD how changes in 25OHD levels relate to changes in lipid levels	Subjects with optimal levels ≥30 ng/mL had lower TC, LDL-C, TGs and higher HDL-C; correcting vitamin D deficiency had no effect on lipids
Li Y. et al. (2021) [28]	USA	Cohort 1: N = 5580 Age: 48 (38–56) yrs 25OHD: 32 ng/mL Cohort 2: N = 6057 Age: 48 (38–56) yrs 25OHD: 34 ng/mL Cohort 3: N = 7249 Age: 49 (39–57) yrs 25OHD: 32 ng/mL	Observational, cross-sectional	lipid profile and 25OHD	Changes in vit D levels correlated negatively with changes in TC, LDL-C, and TGs; no changes in HDL-C levels
Gong T. et al. (2022) [27]	China	153 T2DM ♂ Age: 50.45 ± 11.14 yrs 153 T2DM ♀ Age: 54.14 ± 11.59 yrs	Observational	lipid profile and 25OHD, HOMA-IR, BMI	In overweight/obese with T2DM, serum 25OHD was independently, negatively correlated with TGs
Jorde R. et al. (2010) [114]	Norway	Nonsmokers 8018 Age: 55.9 ± 12.6 yrs 25OHD: 54.1 ± 16.2 nmol/L Smokers 2087 Age: 53.6 ± 11.4 yrs 25OHD: 75.4 ± 20.9 nmol/L Nonsmokers 1762 Age: 55.5 ± 9.8 yrs 25OHD: 55.0 ± 17.7 nmol/L Smokers 397 Age: 52.1 ± 10.3 yrs 25OHD: 68.7 ± 20.8 nmol/L	Cross-sectional Longitudinal	serum 25OHD, TC, HDL-C, LDL-C, LDL-C/HDL-C ratio and triacylglycerol (TAG)	A significative decrease in TGs with the increase in vitamin D
Pathania M. et al. (2023) [29]	India	120 ♂ and 115 ♀ MetS patients Age: 43.81 ± 10.45 yrs 25OHD: 19.14 ± 20.44 ng/mL	Single-center Cross-sectional	serum 25OHD, TC, HDL-C, LDL-C, LDL-C/HDL-C ratio, TGs	Low vit D serum levels show weak correlation with TC, TGs and LDL-C
Atia T. (2023) [32]	Saudi Arabia	n145 non diabetes: 25OHD: 30.28 ± 12.51 ng/mL n104 prediabetes: 25OHD: 24.86 ± 10.59 ng/mL	Cross-sectional study	serum 25OHD, TC, HDL-C, LDL-C, TGs, HOMA-IR, BMI, FG,	Vitamin D deficiency was more prevalent in prediabetes and it was associated with high TG and low HDL levels, with no significant changes in TC or LDL levels
Cheng YL (2023) [31]	Taiwan	118 (53 ♂ 65 ♀; Age, 54.4 ± 10.6 yrs) 25OHD at baseline: 22.7 (17.6–29.2) (ng/mL)	Retrospective study	serum 25OHD, TC, HDL-C, LDL-C, TGs, HbA1c	Increased of 25OHD levels showed a significant reduction in TGs and TC
Chen CW et al. (2023) [33]	Taiwan	407 ♂ and 569 ♀ Age: 20–45 yrs 25OHD < 12 ng/mL N = 205 age: 31.64 ± 4.59 yrs 25OHD 12–200 ng/mL N = 345 age: 31.54 ± 4.34 yrs 25OHD 20–30 ng/mL N = 344 age: 32.35 ± 4.61 yrs 25OHD > 30 N = 82 age: 32.65 ± 4.32 yrs	Single-center Cross-sectional	serum 25OHD, TC, HDL-C, LDL-C, TGs	Vit D deficiency was associated with higher TC, LDL-C, TGs, and non HDL-C

**Abbreviations**: ♂, male; ♀, female; 25OHD, 25-hydroxyvitaminD; y-GT, serum gamma glutamyl transpeptidase; TC, total cholesterol; HDL-C, high-density lipoprotein cholesterol; apo A-1, apolipoprotein A-I; apoB, apolipoprotein B; BMI, bone mass index; FG, fasting glucose; TGs, triglycerides; LDL-C, low-density lipoprotein cholesterol; T2DM, Type 2 diabetes mellitus; uCRP, Ubiquitin Cross-Reactive Protein; IL-6, interleukin 6; IL-8, interleukin 8; VLDL-C, very low-density lipoprotein cholesterol; FPG, fasting plasma glucose; H2PG, 2h postprandial plasma glucose; HbA1c, glycohemoglobin; hsCRP, high-sensitivity C-reactive protein; TNFR-2, Tumor Necrosis Factor Receptor 2; HOMA-IR, Homeostatic Model Assessment of Insulin Resistance; IFG, impaired fasting glucose; PTH, parathyroid hormone; LPL, lipoprotein lipase; FFAs, free fatty acids; MetS, metabolic syndrome.

**Table 2 nutrients-16-01144-t002:** Main characteristics of the interventional studies.

Study/Year	Country	Healthy/ Comorbidity	Subjects	Study	Description	Results
Carlson LA et al. (1970) [57]	Sweden	Healthy	121 ♂ aged 34.6 (range 21–64 yrs) Group A N = 43 Group B N = 32 Group C N = 46	interventional	*Duration of treatment: 6 weeks* (A) No treatment (B) 500 IU vitamin D/day (C) 1000 IU vitamin D/day	vitamin D had no effect on the serum lipid levels
Scragg R et al. (1995) [61]	UK	Healthy	199 subjects Age: 70 yrs (range 63–76) Vit D: N = 95 Placebo: N = 94	randomized double-blind trial	*Duration of treatment:* *a single oral dose of 2.5 mg cholecalciferol* Assessment after 5 weeks	vitamin D supplementation had no effect on serum lipid levels
Andersen R et al. (2009) [62]	Denmark	Healthy	89 ♀–84 ♂ Pakistani immigrants Vit D 10 mcg: N = 56 Vit D 20 mcg: N = 61 Placebo N = 56	1-year-long randomized double-blind placebo-controlled intervention	*Duration of treatment:* *12 months:* Vit D 10 mcg daily Vit D 20 mcg daily	vitamin D had no effect on serum lipid levels
Makariou S et al. (2017) [63]	Greece	MetS subjects	50 MetS subjects Vit D: N = 25 Age: 52 ± 9 yrs 25OHD status: 16.1 ng/mL Placebo: N = 25 Age: 51 ± 12 yrs 25OHD status: 9.9 ng/mL	pilot study PROBE (prospective, randomized, open-label, blinded end-point) design	*Duration of treatment:* *3 months:* Vit D: 2000 IU vitamin D/day	vitamin D had no effect on serum lipid levels
Makariou S et al. (2019) [64]	Greece	MetS subjects	50 MetS subjects Vit D: N = 25 Age: 52 ± 9 yrs 25OHD status: 16.1 ng/mL Placebo: N = 25 Age: 53 ± 7 yrs 25OHD status: 9.9 ng/mL	pre-specified analysis of a previous study	*Duration of treatment:* *3 months:* Vit D: 2000 IU vitamin D/day	vitamin D had no effect on oxidative stress markers
Wongwiwatthananukit S et al. (2013) [65]	USA	MetS subjects	46 ♀–44 ♂ patients with MetS Age: 63.6 ± 11.7 yrs 25-OHD status: 15.19 ± 3.23 ng/mL Placebo: N = 30 DP 20.000 IU: N = 30 DD 40.000 IU: N = 30	a prospective, randomized, double-blind, double-dummy, parallel trial	*Duration of treatment:* *8 weeks:* Group DP: vitamin D 20.000 IU/week, Group DD: vitamin D 40.000 IU/week,	vitamin D had no effect on serum lipid levels
Yin X et al. (2016) [67]	China	MetS subjects	126 MetS subjects Age: 49.5 ± 8.72 yrs 25-OHD status: 14.5 ± 3.3 ng/mL Vit D: N = 61 Placebo: N = 62	1-year long randomized double-blind placebo-controlled	*Duration of treatment:* *1 year:* Vit D: daily 700 IU vitamin D	vitamin D had no effect on serum lipid levels
Farag A et al. (2019) [92]	Iraq	MetS subjects	70 MetS patients Vit D: N = 24 Age: 40.5 ± 5.9 yrs 25-OHD status: 10.7 ± 2.8 ng/mL Vit D + PA: N = 21 Age: 40.4 ± 5.9 yrs 25-OHD status: 10.4 ± 3.2 ng/mL Placebo: N = 25 Age: 42.6 ± 5.6 yrs 25-OHD status: 12.2 ± 3.9 ng/mL	randomized controlled trial	*Duration of treatment:* *12 weeks:* Vit D: 2000 UI/day vitamin D, Vit D + PA: 2000 UI/day vitamin D and physical activity	only vitamin D + PA reduced TC, LDL-C and HDL-C
Imga N N et al. (2019) [101]	Turkey	Overweight/obese	72 ♀ overweight Age: 42.5 ± 10.8 yrs 25OHD status: 6.1 ng/mL 50 ♀ obese Age: 43.9 ± 10.1 yrs 25OHD status= 5.6 ng/mL	interventional study	*Duration of treatment:* *6 months*: 100.000 IU/week as a loading dose for 8 weeks following a maintenance dose of 3000 IU/day	vitamin D reduced LDL-C and HOMA-IR
Patwardhan V. et al. (2017) [91]	India	Healthy	150 healthy Indians Increase Sunlight: N = 50 Age: 47.6 ± 6.6 yrs 25OHD status: 35.6 ± 11.8 nmol/L Vit D: N = 50 Age: 47.5 ± 6.4 yrs 25OHD status: 31.9 ± 12.7 nmol/L Control: N = 50 Age: 47.7 ± 6.8 yrs 25OHD status: 66.3 ± 13.8 nmol/L	prospective, randomized open-label trial	*Duration of treatment:* *6 months:* Increase Sunlight: Sunlight exposure 20 min forearms and face between 11 a.m. and 3 p.m.; Vit D: cholecalciferol 1000 IU/day	sunlight exposure reduced TC, LDL-C, and HDL-C vitamin D supplementation increased TC and HDL-C
Jorde R et al. (2010) [116]	Norway	Overweight/obese	438 overweight or obese subjects Vit DD group: N = 150 Age: 46.3 ± 11.3 yrs 25-OHD status: 57.7 ± 21.2 nmol/L Vit DP group: N = 139 Age: 47.3 ± 11.9 yrs 25-OHD status: 56.7 ± 21.2 nmol/L Placebo group: N = 149 Age: 48.9 ± 11.0 yrs 25-OHD status: 58.8 ± 21 nmol/L	1 year, double-blind placebo-controlled intervention trial	*Duration of treatment:* *1 yrs:* Vit DD: cholecalciferol 40,000 IU per week, Vit DP: cholecalciferol 20,000 IU per week,	vitamin D had no effect on serum lipid levels
Salekzamani S et al. (2016) [93]	Iran	MetS subjects	80 MetS subjects (40.49 ± 5.04 yrs) 25OHD status < 75 nmol/L Vit D group: N = 35 Placebo group: N = 36	randomized, controlled, double-blind study	*Duration of treatment:* *16 weeks:* Vit D group: 50.000 IU vitamin D weekly	vitamin D supplementation decreased TGs and TG/HDL
Heikkinen A M et al. (1997) [60]	Finland	Postmenopausal women in hormone replacement therapy	464 ♀ HRT group N = 65 Age: 52.9 ± 0.29 yrs Vit D N = 83 Age: 52.8 ± 0.24 yrs HRT + Vit D N = 77 Age: 52.4 ± 0.28 yrs Placebo N = 95 Age: 52.5 ± 0.22 yrs	interventional study	*Duration of treatment:* *3 yrs:* HRT (2 mg estradiol valerate + 1 mg cyproterone acetate) Vit D3 (cholecalciferol 300 IU/day) HRT + Vit D3 (both as above), Placebo (calcium lactate 500 mg/day)	vitamin D had no effect on serum lipid levels
Zittermann A. et al. (2009) [114]	Germany	Overweight/obese	200 overweight subjects 25OHD status= 12 ng/mL Vit D group: N = 82 Age: 47.4 ± 10.3 yrs Placebo group: N = 83 Age: 48.8 ± 10.1 yrs	double-blind placebo-controlled intervention trial	*Duration of treatment:* *12 months:* Vit D group: 83.3 mcg/day (3332 IU) cholecalciferol/daily	vitamin D reduced TGs and increased LDL-C
von Hurst P. et al. (2010) [70]	New Zealand	Insulin resistance	81 ♀ insulin resistant 25OHD status < 50 nmol/L Vit D group: N = 42 Age: 41.8 ± 10.1 yrs Placebo: N = 39 Age: 41.5 ± 9.1 yrs	randomized, placebo- controlled, double-blind trial	*Duration of treatment:* *6 months:* Vit D: cholecalciferol 4000 IU/day	vitamin D had no effect on serum lipid levels
Nagpal J. et al. (2009) [66]	India	Healthy	100 ♂ Vit D: N = 35 Age: 42.4 ± 6.6 yrs Control: N = 36 Age: 45.0 ± 9.2 yrs	double-blind randomized controlled trial	*Duration of treatment:* *6 weeks:* Vit D: 3 doses of vitamin D (120,000 IU) fortnightly	vitamin D had no effect on serum lipid levels
Witham M D et al. (2013) [69]	UK	Healthy	50 ♀ Vit D group N = 25 Age: 41.7 ± 13.4 yrs 25OHD status: 27 ± 13 nmol/L Placebo N = 25 Age: 39.4 ± 11.8 yrs 25OHD status: 28.7 ± 5.5 nmol/L	parallel-group, double-blind, randomized placebo-controlled trial.	*Duration of treatment:* *8 week:* Vit D group: a single dose of 100,000 IU vitamin D3	vitamin D had no effect on serum lipid levels
Shab-Bidar S. et al. (2011) [103]	Iran	T2DM	43 ♂–57 ♀ T2DM (52.5 ± 7.4 yrs) PYD group: N = 50 FYD group: N = 50	randomized controlled clinical trial (RCT)	*Duration of treatment:* *12 weeks:* PYD: yogurt drink (170 mg calcium) twice a day; FYD: D-fortified yogurt drink (170 mg calcium + Vit D 500 IU/250 mL) twice a day	vitamin D decreased TGs, TC and LDL-C and increased HDL-C
Nikooyeh B. et al. (2011) [68]	Iran	T2DM	35 ♂–55 ♀ T2DM Age: 50.7 ± 6.1 yrs PY: N = 30 DY: N = 30 DCY: N = 30	interventional study	*Duration of treatment:* *12 weeks:* PY= plain yogurt (150 mg Ca) twice per day; DY = vit D-fortified yogurt drink (500 IU vitamin D3 and 150 mg Ca) twice per day; DCY = calcium + vit D-fortified yogurt drink (500 IU vitamin D3 and 250 mg Ca) twice per day;	vitamin D had no effect on serum lipid levels
Wood A. et al. (2012) [71]	UK	Healthy post menopausal women	305 healthy PMO ♀ Age: 63.8 ± 2.2 yrs Vit D 400 IU: N = 102 25OHD status: 32.7 ± 12.9 nmol/L Vit D 1000 IU: N = 101 25OHD status: 32.4 ± 13.8 nmol/L Placebo: N = 102 25OHD status: 36.2 ± 17.1 nmol/L	parallel-group, double-blind, placebo-controlled randomized controlled trial.	*Duration of treatment:* *1 yrs:* Vit D3 400 UI daily Vit D3 1000 UI daily	vitamin D had no effect on serum lipid levels
Muldowney S. et al. (2012) [73]	Ireland	Healthy	aged 20–40 yrs N = 202 Age: 29.9 ± 6 yrs 25OHD status: 70.4 nmol/L aged > 64 yrs N = 192 Age: 70.8 ± 5 yrs 25OHD status: 54.2 nmol/L	two separate, identical, double-blind, randomized, placebo-controlled intervention studies	*Duration of treatment:* *22 weeks in winter:* All group assumed doses of 0, 5, 10, or 15 mcg daily of cholecalciferol	vitamin D had no effect on serum lipid levels
Chai W. et al. (2013) [72]	Hawaii	Patients with colorectal adenoma	92 colorectal adenoma (30–75 yrs of age) Placebo Group: N = 23 Age: 58.5 ± 21.2 yrs Ca group: N = 23 Age: 61.9 ± 8.2 yrs Vit D group: N = 23 Age: 60.2 ± 8.1 yrs Ca + Vit D group: N = 23 Age: 62.1 ± 7.5 yrs	pilot, randomized, double-blind, placebo-controlled, 2 × 2 factorial design, 6-month clinical trial	*Duration of treatment:* *6 months:* Ca group: calcium carbonate 1 g twice daily Vit D group: cholecalciferol 400 IU twice daily Ca + Vit D group: calcium carbonate 1 g twice daily + 400 IU vitamin D twice daily	vitamin D had no effect on serum lipid levels
Breslavsky A. et al. (2013) [74]	Israel	T2DM	47 T2DM patients Group Vit D N = 24 Age: 66.8 ± 9.2 yrs 25OHD status: 12.91 ± 10.69 ng/mL Group placebo N = 23 Age: 65.8 ± 9.7 yrs 25OHD status: = 10.79 ± 6.57 ng/mL	randomized, double-blind, placebo-controlled study	*Duration of treatment:* *12 months:* Group Vit D: cholecalciferol 1000 IU/day	vitamin D had no effect on serum lipid levels
Sadiya A. at al. (2015) [75]	UAE	Obese and T2DM	87 obese and T2DM Vit D group: N = 45 Age: 49 ± 8 yrs 25OHD status: 28.5 ± 9.2 nmol/L Placebo group: N = 45 Age: 48 ± 8 yrs 25OHD status: 30.5 ± 11.3.2 nmol/L	randomized double-blind clinical trial	*Duration of treatment:* *3 months:* Vit D group 6000 IU vit D/day *For 3 months:* Vit D group 3000 IU vit D/day *For 6 months:* All 2200 IU vit D/day	vitamin D had no effect on serum lipid levels
Moghassemi S et al. (2014) [76]	Iran	Healthy postmenopausal women	76 ♀ healthy PM Vit D group: N = 43 Age: 52.73 ± 4.56 yrs 25OHD status: 34.45 ± 4.9 nmol/L Placebo group: N = 36 Age 51.90 ± 9.94 yrs 25OHD status 33.13 ± 19.77 nmol/L	randomized, double-blind, placebo-controlled, parallel-group study	*Duration of treatment:* *12 weeks:* Vit D3 group: 2000 IU once daily	vitamin D had no effect on serum lipid levels
Dalbeni A. et al. (2014) [78]	Italy	Chronic heart failure patients	23 chronic HF patients Vit D group: N = 13 Age: 71.2 (67.0–75.4) yrs 25OHD status: 16.2 (11.8–20.7) ng/mL Placebo group: N = 10 Age: 73.4 (64.1–82.7) yrs 25OHD status: 16.0 (11.9–20.2) ng/mL	a double-blind, randomized, placebo-controlled trial	*Duration of treatment:* *6 months;* 4000 IU/daily of cholecalciferol	vitamin D had no effect on serum lipid levels
Ryu O. et al. (2014) [79]	Korea	T2DM	62 T2DM subjects Vit D group: N = 40 Age: 54.5 ± 7.4 yrs 25OHD status: 10.7 ± 2.6 ng/mL Placebo group: N = 41 Age: 56.7 ± 7.9 yrs 25OHD status: 12.3 ± 3.0 ng/mL	prospective, randomized, double-blind-ed, placebo-controlled trial	*Duration of treatment:* *24 weeks;* Vit D group: 1000 IU of cholecalciferol + 100 mg calcium twice daily Placebo group: 100 mg calcium twice daily	vitamin D had no effect on serum lipid levels
Yousefi Rad E. et al. (2014) [58]	Iran	T2DM	58 T2DM subjects Vit D group: N = 28 Age: 50.03 yrs 25OHD status: 15.55 ± 1.91 ng/mL Placebo group: N = 30 Age: 49.9 yrs 25OHD status: 14.64 ± 2.22 ng/mL	randomized controlled trial study	*Duration of treatment:* *2 months:* 4000 IU Vitamin D/day	HDL-C level increased significantly in both groups
Kim HJ. et al. (2014) [80]	Korea	T2DM	52 T2DM subjects Vit D + Training group: N = 15 Age: 69.53 ± 0.84 yrs 25OHD status: 11.91 ± 1.66 ng/mL Training group: N = 13 Age: 68.54 ± 1.18 yrs 25OHD status: 13.05 ± 1.43 ng/mL Vit D group: N = 11 Age: 73.27 ± 2.0 6 yrs 25OHD status: 10.44 ± 1.80 ng/mL Control group: N = 13 Age: 70.08 ± 1.37 yrs 25OHD status: 11.66 ± 2.80 ng/mL	interventional study	*Duration of treatment:* *12 weeks:* Vit D + Training group: Vitamin D 1200 IU + exercise 3–4 times/week for 20 min Training group: exercise Vit D group: vitamin D 1200 IU per day	vitamin D combined with exercise training reduced TC, TGs, LDL-C and increase HDL-C
Kampmann U. et al. (2014) [81]	Denmark	T2DM	16 T2DM patients Vit D group: N = 8 Age: 61.6 ± 4.4 yrs 25OHD status: 31.0 ± 4.9 nmol/L Placebo group: N = 8 Age: 57 ± 4.5 yrs 25OHD status: 34.8 ± 3.8 nmol/L	a randomized, placebo-controlled, double-blind trial	*Duration of treatment:* *2 weeks:* Vit D group: cholecalciferol 11,200 IU daily For 10 weeks: Vit D group: cholecalciferol 5600 IU daily	vitamin D had no effect on serum lipid levels
Eftekhari MH et al. (2014) [59]	Iran	T2DM	70 T2DM patients Treatment group: N = 35 Age: 53.8 ± 8.9 yrs Placebo group: N = 35 Age: 52.4 ± 7.8 yrs	double-blind randomized placebo-controlled	*Duration of treatment:* *12 weeks:* Treatment group: Calcitriol 0.25 mcg twice/day	there was a reduction in TC, TGs and LDL-C in all groups; HDL-C only in placebo group
Al-Zahrani MK et al. (2014) [82]	Saudi Arabia	T2DM	200 T2DM patients Vit D group: N = 100 Age: 56.9 ± 9.4 yrs 25OHD status: 25.3 ± 13.8 nmol/L Placebo group: N = 100 Age: 52.5 ± 8.1 yrs 25OHD status: 22.0 ± 15.2 nmol/L	randomized placebo-controlled	*Duration of treatment:* *3 months:* Vit D group: cholecalciferol 45,000 IU orally once every week for 2 months and a single 45,000 I.U. dose in the last month	vitamin D had no effect on serum lipid levels
Asemi Z et al. (2015) [94]	Iran	Overweight/obese women with PCOS	104 ♀ overweight and obese with PCOS Placebo group: N = 26 Age: 24.3 ± 5.2 yrs 25OHD status: 14.0 ± 4.1 ng/mL Calcium group: N = 26 Age 25.0 ± 6.7 yrs 25OHD status: 13.9 ± 2.0 ng/mL Vit D group: N = 26 Age 25.6 ± 4.4 yrs 25OHD status: 11.6 ± 4.7 ng/mL Vit D + Calcium group: N = 26 Age: 24.9 ± 5.1 yrs 25OHD status: 15.1 ± 3.6 ng/mL	randomized double-blind placebo controlled clinical trial	*Duration of treatment:* *8 weeks* Calcium group: 1000 mg/d calcium Vit D group: 50,000 IU/week Vit D + calcium group: 1000 mg calcium/d + 50.000 IU/week	calcium + vitamin D reduced TGs and VLDL-C
Qin XF et al. (2015) [95]	China	Hypercholesterolemia	56 subjects with hypercholesterolemia Vit D group: N = 100 Age: 67.7 ± 8.9 yrs 25OHD status: 21.1 ± 11.7 ng/mL Placebo group: N = 100 Age: 67.8 ± 8.1 yrs 25OHD status: 21.2 ± 11.4 ng/mL	single-center, double-blind, placebo-controlled trial	*Duration of treatment:* *6 months:* Vit D group: add-on to statin vitamin D 2000 IU/day	vitamin D reduced TC, LDL-C and TGs and increase HDL-C
Muñoz-Aguirre P. at al. (2015) [96]	Mexico	T2DM	104 ♀ T2DM Vit D group: N = 52 Age: 56.1 ± 5.1 yrs 25OHD status: 54.8 ± 14.3 nmol/L Placebo group: N = 52 Age: 57.4 ± 5.0 yrs 25OHD status: 54.3 ± 17.1 nmol/L	randomized, double-blind, placebo-controlled,	*Duration of treatment:* *6 months;* Vit D group: 4000 IU vitamin D daily	vitamin D reduced TGs
Jafari T. et al. (2015) [55]	Iran	T2DM	59 ♀ T2DM Vit D fortified yogurt (FY): N = 30 Age: 57.8 ± 5.5 yrs 25OHD status: 62.23 ± 4.52 nmol/L plain yogurt (PY): N = 29 Age: 56.8 ± 5.7 yrs 25OHD status: 62.72 ± 4.27 nmol/L	double-blind randomized placebo-controlled trial	*Duration of treatment:* *12 weeks:* FY: 2000 IU vitamin D in 100 g/day	vitamin D had no effect on serum lipid levels
Riek AE et al. (2018) [109]	USA	T2DM	26 T2DM patients 25OHD: 17 ng/mL Vit D group: N = 11 Age: 57.6 ± 1.9 yrs Placebo group: N = 15 Age: 57.4 ± 1.8 yrs	interventional study	*Duration of treatment:* *4 months:* Vit D group: vitamin D 4000 IU/day	vitamin D had no effect in plasma lipid (vitamin D reduced total monocyte cholesterol content by suppressing oxidized LDL cholesterol uptake)
Liyanage GC. et al. (2017) [106]	Sri Lanka	T2DM + early stage nephropathy	85 T2DM + early stage nephropathy patients Vit D3 group: N = 42 Age: 56 ± 10 yrs 25OHD status: 55.9 ± 12.3 nmol/L Placebo group: N = 43 Age: 59 ± 8 yrs 25OHD status: 50.0 ± 9 nmol/L	randomized double-blind clinical trial	*Duration of treatment:* *6 months:* 50.000 UI vitamin D/month intramuscularly	vitamin D reduce TC, LDL-C and increase HDL-C
Jamilian M. et al. (2017) [97]	Iran	Gestational diabetes	140 ♀ GDM Placebo: N = 35 Age: 30.7 ± 4.1 yrs 25OHD status: 16.6 ± 2.6 ng/mL Vit D: N = 35 Age: 31.5 ± 7.0 yrs 25OHD status: 15.2 ± 3.8 ng/mL Omega-3: N = 35 Age: 30.7 ± 3.5 yrs 25OHD status: 16.9 ± 3.5 ng/mL Vit D + Omega-3: N = 35 Age: 31.2 ± 4.3 yrs 25OHD status: 15.5 ± 3.1 ng/mL	randomized double-blind placebo-controlled clinical trial	*Duration of treatment:* *6 weeks:* Vit D: 50,000 IU vitamin D every 2 weeks Omega-3: 1000 mg omega 3 twice a day Vit D + Omega-3: 50,000 IU vitamin D every 2 weeks + 1000 mg omega 3 twice a day	in the group with vit D + omega 3 co-supplementation there was a reduction in TGs
Ghaderi A. et al. (2017) [98]	Iran	Patients in maintenance methadone treatment	68 MMT subjects Vit D3 group N = 34 Age: 42.5 ± 8.9 yrs 25OHD status: 13.9 ± 4.5 ng/mL Placebo group N = 34 Age: 40.1 ± 9.2 yrs 25OHD status: 13.5 ± 4.5 ng/mL	randomized, double-blind, placebo-controlled, clinical trial	*Duration of treatment:* *12 weeks:* Vit D: 50,000 IU vitamin D every 2 weeks	vitamin D supplementation reduced TC, TGs, LDL-C and increased HDL-C
Ghaderi A. et al. (2019) [99]	Iran	Chronic schizophrenia patients	60 chronic schizophrenia patients Vit D3 group N = 30 Age: 44.8 ± 8.3 yrs 25OHD status: 15.0 ± 4.1 ng/mL Placebo group N = 30 Age: 43.2 ± 6.0 yrs 25OHD status: 13.7 ± 3.2 ng/mL	randomized, double-blind, placebo-controlled, clinical trial	*Duration of treatment:* *12 weeks:* Vit D: 50,000 IU vitamin D + probiotic every 2 weeks Placebo: probiotic	vitamin D + probiotic reduced TGs, TC, LDL-C and VLDL-C
Tamadon M. et al. (2018) [77]	Iran	Diabetic patients on hemodialysis	60 diabetic hemodialysis patients Vit D group: N = 30 Placebo group: N = 30	randomized, double-blind, placebo-controlled, clinical trial	*Duration of treatment:* *12 weeks:* Vit D: 50.000 UI every 2 weeks;	vitamin D had no effect on serum lipid levels
Yarparvar A. et al. (2020) [43]	Austria	Healthy	71 healthy adolescent boys (age 17 yrs) Vit D group: N = 34 25OHD status: 24.16 ± 10.16 ng/mL Placebo group: N = 37 25OHD status: 22.15 ± 12.33 ng/mL	randomized single-blind placebo-controlled trial	*Duration of treatment:* *6 months:* Vit D: 50.000 IU monthly	vitamin D reduced TC and LDL-C
Schwetz V. at al. (2018) [113]	Austria	Hypertension	163 subjects with hypertension Vit D3 group N = 79 Age: 62.2 yrs 25OHD status: 52.5 nmol/L Placebo group N = 84 Age: 62.1 yrs 25OHD status: 58.8 nmol/L	single-center, double-blind, randomized, placebo-controlled	*Duration of treatment:* *8 weeks:* Vit D group: 2800 IU of vitamin D daily	vitamin D increased TC and TGs
Hafez M. at al. (2019) [83]	Egypt	T1DM children	50 children with T1DM (for >2 yrs) and dyslipidemia VD sufficiency (>30 ng/mL) N = 20 VD deficiency (VDD) or insufficiency (<29 ng/mL) N = 30	prospective cohort study	*Duration of treatment:* *4 months:* VDD: 4000 UI/day of vitamin D	vitamin D had no effect on serum lipid levels
Mohamad M.at al. (2016) [84]	Egypt	T2DM	100 T2DM patients Age: 47.35 ± 6 25OHD status: 16 ± 5.3 ng/mL	interventional study	*Duration of treatment:* *2 months:* 4500 IU/day of vitamin D	vitamin D had no effect on serum lipid levels
Miao J. et al. (2021) [100]	USA	Hypertension	289 subjects at high risk of hypertension Age: 37.0 yrs 25OHD status: 15.1 ng/mL Low-dose Vit D: N = 144 High-dose Vit D: N = 145	randomized, double-blind, controlled trial	*Duration of treatment:* *6 months:* Low-dose Vit D: 400 IU/daily vitamin D; High-dose Vit D: 4000 IU/daily vitamin D	vitamin D increased TGs in high-dose vitamin D group
Jastrzebski Z.at al. (2016) [85]	Poland	Healthy	16 professional rowers Vit D group N = 8 Age: 23.1 ± 2.7 yrs 25OHD status: 35.7 ± 17.0 nmol/L Placebo group N = 8 Age: 23.1 ± 3.2 yrs 25OHD status: 31.4 ± 15.2 nmol/L	interventional study	*Duration of treatment:* *4 weeks:* Vit D: 5.000 IU of vitamin D every day	vitamin D had no effect on serum lipid levels
Khosravi ZS et al. (2018) [86]	Iran	Overweight/obese	53 ♀ overweight and obese Intervention N = 26 Age: 29.1 ± 9.6 yrs 25OHD status: 22 ± 6.5 nmol/L Placebo N = 27 Age: 26.9 ± 9.1 yrs 25OHD status: 18 ± 5 nmol/L	double-blind clinical trial study	*Duration of treatment:* *6 weeks:* Intervention: 50.000 IU/week	vitamin D had no effect on serum lipid levels
Ramiro-Lozano JM et al. (2015) [104]	Spain	T2DM	28 T2DM Age: 71.7 ± 9.6 25OHD status: 10.6 ± 3.6	interventional study	*Duration of treatment:* *8 weeks:* 16.000 IU of calcifediol orally once a week	vitamin D decreased TC and LDL-C
Hu J. et al. (2018) [111]	China	Mild cognitive impairment	181 subjects with mild cognitive impairment Vit D N = 93 Age: 67.22 ± 6.09 yrs 25OHD status: 19.07 ± 2.91 ng/mL Placebo N = 88 Age: 66.60 ± 5.24 yrs 25OHD status: 19.77 ± 2.91 ng/mL	randomized, double-blind, placebo-controlled trial	*Duration of treatment:* *12 months:* Vit D: 400 UI/day vitamin D	vitamin D reduced TC, TGs, HDL-C and LDL-C
Angellotti E. et al. (2019) [88]	USA	T2DM	127 T2DM patients Vit D N = 66 Age: 60.1 ± 8.4 yrs 25OHD status: 26.6 ± 11.1 ng/mL Placebo N = 61 Age: 60.1 ± 8.1 yrs 25OHD = 25.8 ± 10.3 ng/mL	double-blind, randomized, placebo-controlled clinical trial	*Duration of treatment:* *48 weeks:* Vit D: 4000 IU/day of vit D	vitamin D had no effect on serum lipid levels
Al-Daghri NM et al. (2012) [101]	Saudi Arabia	T2DM	34 ♂ 58 ♀ T2DM patients Vit D levels baseline: 32.2 ± 1.5 nmol/L	multi-center, interventional study	*Duration of treatment:* *18 months:* 2000 IU vitamin D daily	vitamin D supplementation reduced TC and LDL-C and increased HOMA-IR
Dutta D. et al. (2014) [87]	India	Prediabetes and diabetes	121 Prediabetic and diabetic subjects Group A: N = 68 Age: 48.37 ± 10.47 yrs 25OHD status: 17.04 ± 7.66 ng/mL Group B: N = 57 Age: 47.4 ± 11.51 yrs 25OHD status: 18 ± 7.16 ng/mL Group C: N = 45 Age: 46.6 ± 11.01 yrs 25OHD status: 37.89 ± 8.26 ng/mL	interventional study	*Duration of treatment:* *12 months:* Group A: cholecalciferol 60,000 UI/once weekly for 8 weeks than monthly for 12 months + calcium carbonate 1250 mg/day; Group B: calcium carbonate Group C: calcium carbonate All: lifestyle interventions	vitamin D supplementation in prediabetes and diabetes is associated with an improvement in dyslipidemia
Bhatt SP et al. (2020) [89]	India	Prediabetes	121 ♀ prediabetes Aged 20–60 yrs Intervention N = 61 25OHD status: 29.9 ± 5.9 nmol/L Placebo N = 60 25OHD status: 32.1 ± 5.2 nmol/L	open-label randomized placebo-controlled trial	*Duration of treatment:* *8 weeks:* Intervention: cholecalciferol 60,000 IU once per week	vitamin D had no effect on serum lipid levels
Jorde R. et al. (2016) [116]	Norway	Prediabetes	511 subjects with prediabetes Vit D N = 256 Age: 62.3 ± 8.1 yrs 25OHD status: 24 ± 8.8 ng/mL Placebo N = 255 Age: 61.9 ± 9.2 yrs 25OHD status: 24.4 ± 8.5 ng/mL	randomized controlled trial	*Duration of treatment:* *5 yrs:* Vit D: cholecalciferol 20,000 IU weekly for	in the vit D group, there was a decrease in LDL-C only at 1 year
Rajabi-Naeeni M. et al. (2020) [107]	Iran	Prediabetes	168 ♀ prediabetes Placebo: N = 42 Age: 41.8 ± 7.8 yrs 25OHD status: 25.5 ± 5.8 ng/mL Omega-3: N = 42 Age: 39.8 ± 6.9 yrs 25OHD status: 23.6 ± 6.4 ng/mL Vit D: N = 42 Age: 39.9 ± 6.0 yrs 25OHD status: 21.4 ± 8.0 ng/mL Vit D + Omega-3: N = 42 Age: 39.0 ± 7.7 yrs 25OHD status: 22.0 ± 8.9 ng/mL	factorial, triple-blind clinical trial	*Duration of treatment:* *8 weeks:* Omega-3: 1000 mg omega 3 twice daily Vit D: cholecalciferol 50,000 IU every 2 weeks Vit D + Omega-3: 1000 mg omega 3 twice daily + cholecalciferol 50,000 IU every 2 weeks	TC, TGs and LDL-C decrease and HDL-C increase in the vit D + omega 3 group TC and LDL-C decrease in Vit D group
Misra P. et al. (2021) [90]	India	Prediabetes	132 ♀ prediabetes Intervention N = 67 Age: 48.1 ± 6.7 yrs 25OHD status: 27.6 ± 6.5 ng/mL Placebo N = 65 Age: 46.1 ± 8.1 yrs 25OHD status: 27.7 ± 14.6 ng/mL	open-label randomized placebo-controlled trial	*Duration of treatment:* *8 weeks:* Intervention: cholecalciferol 60,000 IU + calcium carbonate 1 gr Placebo: calcium carbonate 1 gr	vitamin D had no effect on serum lipid levels
Sacheck J. et al. (2022) [105]	USA	Healthy children	604 children (8–15 yrs) 600 UI group: N = 207 25OHD status: 21.9 ± 0.5 ng/mL 1000 UI group: N = 190 25OHD status: 21.7 ± 0.5 ng/mL 2000 UI group: N = 207 25OHD status: 22.4 ± 0.5 ng/mL	a randomized double-blind clinical trial	*Duration of treatment:* *6 months:* 600 IU or 1000 IU or 2000 IU daily	vitamin D at low dosage increased HDL-C and at high dosage decreased LDL-C and TC
Samimi M et al. (2015) [108]	Iran	Pregnant women at risk of pre-eclampsia	60 ♀ pregnancy at risk of risk for pre-eclampsia Vit D group: N = 30 Age: 27.3 ± 3.7 yrs 25OHD status: 13.1 ± 6.4 ng/mL Placebo: N = 30 Age: 27.1 ± 5.2 yrs 25OHD status: 16.2 ± 3.5 ng/mL	prospective, double-blind, placebo-controlled trial	*Duration of treatment:* *20 to 30 weeks of gestation*: Vit D group: 50,000 IU cholecalciferol every 2 weeks + calcium 1000 mg day	vit D + calcium increased serum HDL-C
Sabico S et al. (2023) [112]	Saudi Arabia	Healthy	58 ♂ 62 ♀ healthy adults age 40.6 ± 10.8 yrs 25OHD status < 50 nmol/L	interventional study	*Duration of treatment:* *6 months:* 50,000 IU cholecalciferol weekly once for first 2 months, then once in two weeks for month 3 and 4, and a daily dose of 1000 IU for the last two months	vitamin D increased HDL-C and reduced ASCVD risk score
Bahramian H et al. (2023) [117]	Iran	PCOS	80 PCOS ♀ vit D < 20 ng/mL Vit D N = 18 Age 23.6 ± 3.4 yrs Omega 3 N = 20 Age 22.3 ± 3.6 yrs Vit D + Omega-3 N = 20 Age 24.6 ± 3.1 yrs Placebo N = 18 Age 22.3 ± 3.1 yrs	double-blind, randomized clinical trial	*Duration of treatment:* *6 weeks* 50,000 IU/weekly Omega-3	vitamin D + Omega-3 treatment reduce TC
Safari S et al. (2023) [118]	Iran	subclinical hypothyroidism	44 ♀ subclinical hypothyroidism Vit D N = 22 Age 36.3 ± 11.6 yrs placebo N = 22 Age 36.0 ± 11.125 yrs	randomized, double-blind, placebo-controlled clinical trial	*Duration of treatment:* *12 weeks* 50,000 IU/week of vitamin D	vitamin D reduce TC
Ku CW et al. (2024) [119]		Overweight/obese pregnant women	Intervention N = 137 Age: 30.5 ± 4.6 yrs 25OHD = 39.9 4 ± 17.0 nmol/L Control N = 137 Age: 30.6 ± 4.2 yrs 25OHD = 38.9 ± 11.97 nmol/L	two-arm, parallel, non-blind randomized controlled trial	*Duration of treatment:* *from week 16 of gestation to delivery* Intervention: 800 UI/daily of Vit D Control: 400 UI/daily of Vit D	vitamin D had no effect on serum lipid levels

**Abbreviations**: ♂, male; IU, international unit; ♀, female; 25OHD, 25-hydroxyvitamin D; MetS, metabolic syndrome; TC, total cholesterol; TGs, triglycerides; HDL-C, high-density lipoprotein cholesterol; LDL-C, low-density lipoprotein cholesterol; PA, physical activity; HOMA-IR, Homeostatic Model Assessment of Insulin Resistance; HRT, hormone replacement therapy; T2DM, type 2 diabetes mellitus; VLDL-C, very-low-density lipoprotein cholesterol; GDM, gestational diabetes; MMT, maintenance methadone treatment; HD, diabetic hemodialysis patients; ASCVD risk score, 10-year risk of atherosclerotic cardiovascular disease; PCOS, polycystic ovary syndrome.

With regard to the effect on individual classes of the lipid profile, we observed that in patients with T2DM, the most frequent variation concerns TC and LDL-C, while in studies with MetS and overweight (OW) subjects, the effect on the reduction in TGs prevails.

## 4. Discussion

The relationship between vitamin D and cardiovascular risk is very popular today. In most studies, the conclusions are quite consistent: observational studies have shown inverse associations between vitamin D concentration and cardiovascular disease, but the same results are not always replicable in intervention studies. Considering these data, which unfortunately often accompany vitamin D studies, we conducted our analysis.

From the review of the literature on vitamin D and dyslipidemia, the same paradox clearly emerges: while observational studies give mostly unambiguous results and support an evident association between increased serum vitamin D levels and a favorable lipid profile, interventional studies show contrasting data or no effect. So, it would seem that hypovitaminosis D could predispose to dyslipidemia rather than replenishment leading to a significant improvement in lipid levels. Observational studies do not allow us to find a cause-and-effect link between vitamin D and dyslipidemia; however, from the consistent results, it would seem, as in many other fields, that vitamin D deficiency represents a risk marker—in this case, a possible marker of cardiovascular risk. Similarly, inconclusive results have been obtained in several studies involving vitamin D treatment for patients with diabetes [120] or COVID-19 infection [121].

The interventional studies we considered in our review exhibit significant heterogeneity regarding population characteristics (such as age and pathologies) as well as doses, methods of administration, and duration of vitamin D intervention. Nevertheless, upon analyzing these studies, it does not seem that age influenced the results. Surprisingly, positive effects of vitamin D supplementation on lipids were observed across various age groups, including adults (aged over 60) and younger patients, including children.

In most studies where vitamin D supplementation was administered to individuals with vitamin D deficiency, beneficial effects on lipid profiles were observed. Interestingly, the dosage of vitamin D supplementation does not appear to significantly impact these effects. However, there remains a critical gap in knowledge regarding the optimal dose of vitamin D for pleiotropic effects. Discrepancies in study outcomes may also stem from variations in the methods used to assess vitamin D levels; some studies employed validated schemes for assessment, while others did not specify their methods. This discrepancy represents a notable limitation. Indeed, intervention studies with meticulous vitamin D assessment tend to demonstrate positive effects on lipid profiles as a result of vitamin D supplementation.

Some of the most interesting data that emerge from this review show that supplementation with vitamin D is particularly effective in improving and the lipid profile in patients with T2DM, MetS, or OW.

Therefore, diabetic and metabolic patients would have a greater advantage, compared to healthy subjects, in the administration of vitamin D in terms of lipids. This positive effect of vitamin D on dyslipidemia in these patients could be explained by the action of vitamin D on improving insulin resistance (IR). Insulin resistance is an important metabolic component in metabolic syndrome, prediabetes, PCOS, and T2DM and is associated with risk of micro- and macrovascular complications. Insulin resistance is also involved in lipid metabolism, leading to atherogenic dyslipidemia; therefore, vitamin D, by improving IR, would probably have an indirect effect on lipids. Indirectly, also through the control of inflammation, vitamin D could lead to action on lipids. A recent review showed how the administration of vitamin D to patients with T2DM determines a significant reduction in inflammatory markers (CRP and IL-6), with a positive effect on the lipid profile [122].

Similar to other hormones, vitamin D supplementation is beneficial for individuals with deficiency but may not be necessary for everyone. This phenomenon is also evident in addressing bone fragility: while vitamin D supplementation significantly reduces fragility fractures in individuals who are deficient and at risk, administering high doses to healthy individuals does not show a significant effect in preventing such fractures.

Furthermore, the inconsistent outcomes of interventional studies might also stem from variations in dosage, frequency, or duration of vitamin D supplementation. Also, other variables could influence the results of interventional studies, such as physical activity, obesity, VDR polymorphisms, or the heterogeneity of populations. Let us not forget that vitamin D is partly taken with food; this component was not considered in interventional works. Furthermore, cholesterol also depends on diet as well as physical activity. So, in order to carry out more truthful studies, all of these components should be taken into consideration, as they could influence the results of interventional studies.

## 5. Conclusions

The prospect of vitamin D supplementation contributing to the improvement of lipid parameters and reducing cardiovascular risk is truly fascinating. Vitamin D is readily available and generally without side effects when taken in recommended doses. The possibility that its replenishment could play a role in cardiovascular risk is very intriguing and warrants further analysis.

Currently, data in the literature confirm that, at the molecular level, there is a close link between lipid metabolism and vitamin D; numerous observational studies suggest a possible association between vitamin D and serum lipid levels, but the data do not appear to be completely confirmed in interventional studies. However, it was certainly interesting to observe how, in interventional studies on metabolic and diabetic patients, the effects of vitamin D replenishment were positive on lipids.

In the future, this relationship should be further explored in large, well-designed, randomized controlled trials (RCTs). The results of such studies could be truly surprising and could help us better understand the role of vitamin D in cardiovascular prevention.

## Figures and Tables

**Figure 1 nutrients-16-01144-f001:**
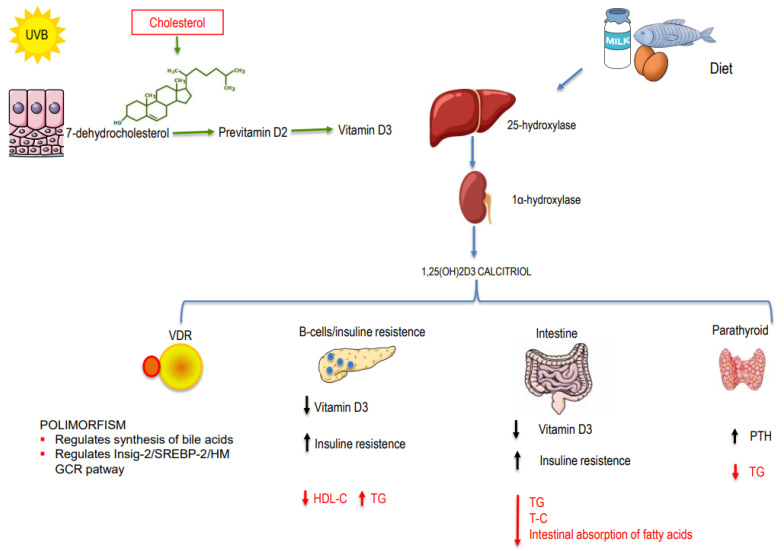
Mechanisms involved in the relationship between vitamin D and dyslipidemia.

**Figure 2 nutrients-16-01144-f002:**
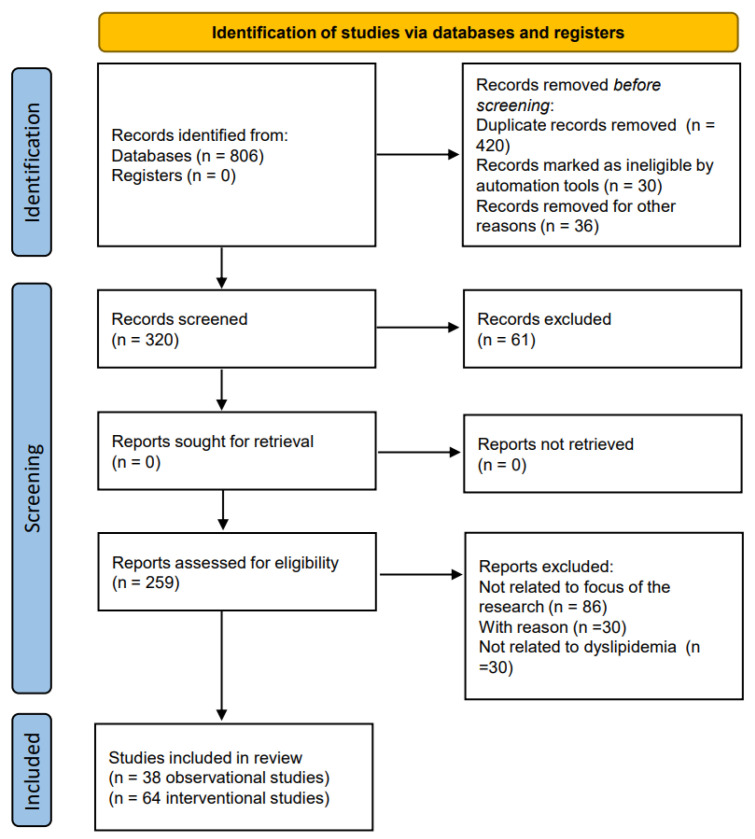
Flow chart of the studies identified and included in this review.

**Figure 3 nutrients-16-01144-f003:**
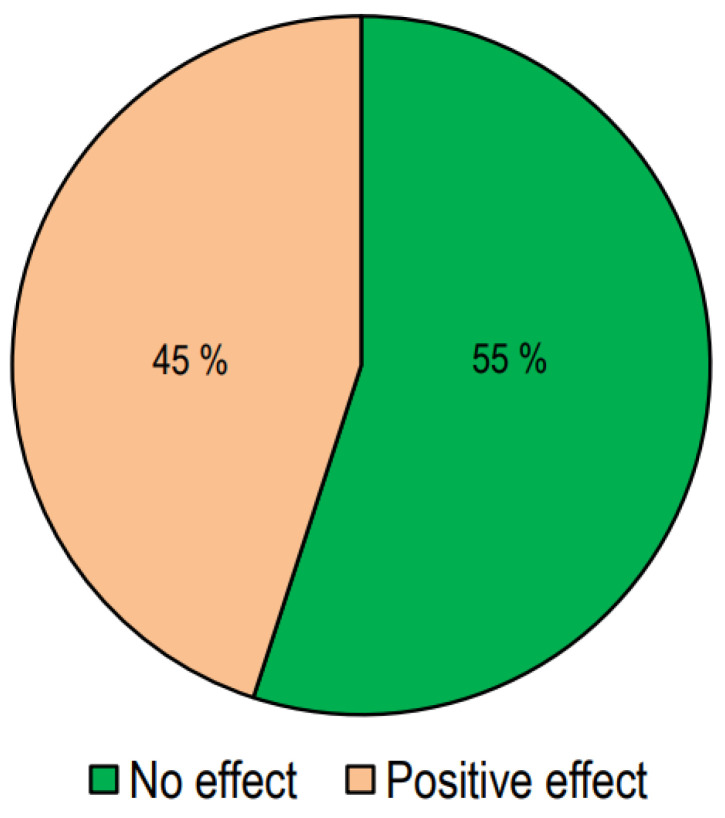
The effect of vitamin D supplementation on lipid profiles in interventional studies.

## Data Availability

No new data were created.

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
