# Peer review of "Vitamin D and Dyslipidemia: Is There Really a Link? A Narrative Review"

_nutrients, 2024, doi:10.3390/nu16081144_

Round 1

Reviewer 1 Report

Comments and Suggestions for Authors

Refaie et. Al. present a review in which Vitamin D and dyslipidemia are considered together. Taken together, the review lacks many key details in honing in on a relationship between the two and in forming a cohesive hypothesis on that relationship. For instance, in the interventional studies where effects of Vitamin D on cholesterol were not observed, where their key details in the ones that were successful (physical exercise, sunlight, dosage amount). Even a summary statement stating how many were positive or negative would be enlightening. Often, an entire study is reduced to a single summary statement describing an effect but no details. Key details are omitted from the introduction in describing the components of cholesterol and how the studies alter each one. Also, there are substantial grammatical and spelling errors throughout the entire document, thus it would be recommended to have a native English speaker review.

Minor:

1.      Abstract, line 1 has grammatical error “extra”. Also incorrectly spelled work “significative”.

2.      25OHD is the circulating form of Vitamin D found in serum correct? That should be clarified in the introduction.

3.      Grammatical error line 23 –“inflammation, immunity, endocrine system”.

4.      Line 24 - It’s mentioned that “several studies have found an association” but only 1 citation is listed.

5.      “Not genetic mechanisms” probably should be more appropriately labeled.

6.      Line 50 states “many papers have showed” but only one is cited.

7.       Line 105 and 107 – I believe the abbreviation is (T2DM).

8.      Table 2 typo “vitamin D had not effect”

9.      Given that obesity is quite common in individuals with dyslipidemia, was this accounted for in your reviews? It would seem somewhat intuitive that an obese individual, especially morbidly obese, would be outside less and exercise less.

1. Figure 2 has the words in the blue boxes two letters at time. It’s essentially unreadable like that, consider placing the words sideways, like a y-axis.

Comments on the Quality of English Language

Needs extensive revision. 

Author Response

Refaie et. Al. present a review in which Vitamin D and dyslipidemia are considered together. Taken together, the review lacks many key details in honing in on a relationship between the two and in forming a cohesive hypothesis on that relationship. For instance, in the interventional studies where effects of Vitamin D on cholesterol were not observed, where their key details in the ones that were successful (physical exercise, sunlight, dosage amount). Even a summary statement stating how many were positive or negative would be enlightening. Often, an entire study is reduced to a single summary statement describing an effect but no details. Key details are omitted from the introduction in describing the components of cholesterol and how the studies alter each one. Also, there are substantial grammatical and spelling errors throughout the entire document, thus it would be recommended to have a native English speaker review.

First of all we want to thank the reviewer for his suggestion. In the review we evaluated separately the effect of supplementation in different groups. We added two figures and discussed the effect of Vitamin D on the lipid profile in the discussion section. 

Minor:

  1. Abstract, line 1 has grammatical error “extra”. Also incorrectly spelled work “significative”.

According to the suggestion of the reviewer we corrected the sentence in abstract

  1. 25OHD is the circulating form of Vitamin D found in serum correct? That should be clarified in the introduction.

I confirm that 25OHD is major circulating form of the Vitamin D

  1. Grammatical error line 23 –“inflammation, immunity, endocrine system”.

According to the suggestion of the reviewer we corrected the grammatical error.

  1. Line 24 - It’s mentioned that “several studies have found an association” but only 1 citation is listed.

According to the suggestion of the reviewer we corrected the sentence

  1. “Not genetic mechanisms” probably should be more appropriately labeled.

According to the suggestion of the reviewer we changed the sentence

  1. Line 50 states “many papers have showed” but only one is cited.

According to the suggestion of the reviewer we corrected the sentence

  1. Line 105 and 107 – I believe the abbreviation is (T2DM).

According to the suggestion of the reviewer we corrected

  1. Table 2 typo “vitamin D had not effect”

According to the suggestion of the reviewer we corrected the sentence

  1. Given that obesity is quite common in individuals with dyslipidemia, was this accounted for in your reviews? It would seem somewhat intuitive that an obese individual, especially morbidly obese, would be outside less and exercise less.

The reviewer’s suggestion is appropriate and in the conclusions we stressed the need for studies that also take into account the influence of lifestyle and physical activity.

  1. Figure 2 has the words in the blue boxes two letters at time. It’s essentially unreadable like that, consider placing the words sideways, like a y-axis.

According to the suggestion of the reviewer Figure 2 has been corrected

Reviewer 2 Report

Comments and Suggestions for Authors

The manuscript explores the relationship between vitamin D status and lipid imbalance (dislipidemia), itself a risk factor for cardiovascular disease, explored by way of a literature review, and presented as a qualitative discussion rather than quantitatively. Since the outcome is discursive it is particularly important that it is clear an unambiguous as there are not figures or graphs of results that a reader can interpret. Some work is needed here.

There are some interesting points in the Discussion and Conclusion that are certainly useful. The possible explanation of differences between observation and intervention studies has been addressed before but bears repeating. The point that effect appears independent of age is interesting. The impact on those already with metabolic syndrome (but not healthy subjects) is also worthy of further confirmation.

Presentation needs a lot of work before publication. A few specific points independent of language are given below.

Figure 1, suspect text size will be far too small on a printed page.

Figure 2, place text in blue boxes vertically (along the length of the boxes). There is no key for * and **, though it looks as though the bottom of the figure has gone off the bottom of the page. Figure captions are usually beneath the figure (check journal instructions).

First line of results – identify reference numbers for the observational and intervention studies you found. There were no meta-analyses and reviews (according to Figure 2). If there are some but you have not used them then you need to clarify.

Author contributions show only the journal explanation of this section

Comments on the Quality of English Language

The manuscript is generally comprehensible, but contains a great deal of poor grammar, odd words and phraseology. It requires extensive language editing. There is a concern that because the subject matter is complex there may be misunderstanding of the results discussed. For example, there is a difference between positive correlation (understood as both variables move up, or down, together), and a positive effect, which could imply the same thing, or the opposite if a positive health benefit is for a variable to decrease. Since the language in general is imprecise, it reduces confidence in understanding exactly what the authors mean with some of their comments.

Author Response

The manuscript explores the relationship between vitamin D status and lipid imbalance (dislipidemia), itself a risk factor for cardiovascular disease, explored by way of a literature review, and presented as a qualitative discussion rather than quantitatively. Since the outcome is discursive it is particularly important that it is clear and unambiguous as there are not figures or graphs of results that a reader can interpret. Some work is needed here.  There are some interesting points in the Discussion and Conclusion that are certainly useful. The possible explanation of differences between observation and intervention studies has been addressed before but bears repeating. The point that effect appears independent of age is interesting. The impact on those already with metabolic syndrome (but not healthy subjects) is also worthy of further confirmation.

First of all we want to thank the reviewer for his suggestion. In the review we evaluated separately the effect of supplementation in different groups. We added two figures and discussed the effect of Vitamin D on the lipid profile in the discussion section. 

Presentation needs a lot of work before publication. A few specific points independent of language are given below.

Figure 1, suspect text size will be far too small on a printed page.

According to the suggestion of the reviewer test size has been enlarged

Figure 2, place text in blue boxes vertically (along the length of the boxes). There is no key for * and **, though it looks as though the bottom of the figure has gone off the bottom of the page. Figure captions are usually beneath the figure (check journal instructions).

      According to the suggestion of the reviewer Figure 2 has been corrected

First line of results – identify reference numbers for the observational and intervention studies you found. There were no meta-analyses and reviews (according to Figure 2). If there are some but you have not used them then you need to clarify.

We selected only observational and interventional studies

Author contributions show only the journal explanation of  this section

The manuscript is generally comprehensible, but contains a great deal of poor grammar, odd words and phraseology. It requires extensive language editing. There is a concern that because the subject matter is complex there may be misunderstanding of the results discussed. For example, there is a difference between positive correlation (understood as both variables move up, or down, together), and a positive effect, which could imply the same thing, or the opposite if a positive health benefit is for a variable to decrease. Since the language in general is imprecise, it reduces confidence in understanding exactly what the authors mean with some of their comments.

The manuscript has been by an experienced English-speaking colleague

Round 2

Reviewer 1 Report

Comments and Suggestions for Authors

Overall, the review is improved. But there are still significant grammatical errors throughout and it really needs to be edited by a native speaker. Other comments are below.

1.      Is Vitamin D also the same as 25OHD? Or is 25OHD metabolites of Vitamin D that can be easily measured in serum? The introduction says they are the same, are they the same?

2.      Grammar is still incorrect in Line 32.

3.      Table 1 is 5 pages and Table 2 is 12 pages. The written review is 26 pages, meaning that That’s almost half the review and is simply too large and too complicated for a reader to glean insight from. It needs to be rearranged or redistributed in a manner that is useful for the reader. Or moved to supplement or somehow minimized.

4.       The term OW is not defined.

Comments on the Quality of English Language

Needs significant improvement. 

Author Response

  1. Is Vitamin D also the same as 25OHD? Or is 25OHD metabolites of Vitamin D that can be easily measured in serum? The introduction says they are the same, are they the same?

According to the suggestion of the Reviewer, we have improved in the introduction section the function of Vitamin D.  Of course we consider 25OHD that measured in serum as is commonly carried out in clinical practice

  1. Grammar is still incorrect in Line 32.

In this version, we have revised the English language using the support of an expert mother tongue

  1. 3.Table 1 is 5 pages and Table 2 is 12 pages. The written review is 26 pages, meaning that That’s almost half the review and is simply too large and too complicated for a reader to glean insight from. It needs to be rearranged or redistributed in a manner that is useful for the reader. Or moved to supplement or somehow minimized.

taking the suggestion we asked the magazine if we can expose the tables more succinctly

  1. The term OW is not defined.

OW term was added

Reviewer 2 Report

Comments and Suggestions for Authors

Thank you for the improvements made to this manuscript. The new figures are simple but helpful, and the new section in the Discussion is also useful.

Not everything has been addressed, for example on line 76 you still say a range of types of study are available and do not clarify that you have excluded meta-analyses and reviews (otherwise why mention them?).

Comments on the Quality of English Language

There is still rather a lot of minor language editing that is required and this would be a job for the journal to address in detail, not a reviewer. I therefore leave it to the Editors to determine whether the language is of a suitable standard (or can be corrected in the preparation of the final proofs).

Author Response

Not everything has been addressed, for example on line 76 you still say a range of types of study are available and do not clarify that you have excluded meta-analyses and reviews (otherwise why mention them?).

According to the suggestion of the studies we change the statement on line 76

There is still rather a lot of minor language editing that is required and this would be a job for the journal to address in detail, not a reviewer. I therefore leave it to the Editors to determine whether the language is of a suitable standard (or can be corrected in the preparation of the final proofs).

In this version, we have revised the English language using the support of an expert mother tongue